# Polyphenol Compound 18a Modulates UCP1-Dependent Thermogenesis to Counteract Obesity

**DOI:** 10.3390/biom14060618

**Published:** 2024-05-23

**Authors:** Xueping Wen, Yufei Song, Mei Zhang, Yiping Kang, Dandan Chen, Hui Ma, Fajun Nan, Yanan Duan, Jingya Li

**Affiliations:** 1School of Chinese Materia Medica, Nanjing University of Chinese Medicine, Nanjing 210023, China; 2Shanghai Institute of Materia Medica, Chinese Academy of Sciences, 189 Guo Shou Jing Road, Shanghai 201203, Chinafjnan@simm.ac.cn (F.N.)

**Keywords:** 18a, adipocytes, polyphenols, thermogenesis, UCP1

## Abstract

Recent studies increasingly suggest that targeting brown/beige adipose tissues to enhance energy expenditure offers a novel therapeutic approach for treating metabolic diseases. Brown/beige adipocytes exhibit elevated expression of uncoupling protein 1 (UCP1), which is a thermogenic protein that efficiently converts energy into heat, particularly in response to cold stimulation. Polyphenols possess potential anti-obesity properties, but their pharmacological effects are limited by their bioavailability and distribution within tissue. This study discovered **18a**, a polyphenol compound with a favorable distribution within adipose tissues, which transcriptionally activates UCP1, thereby promoting thermogenesis and enhancing mitochondrial respiration in brown adipocytes. Furthermore, in vivo studies demonstrated that **18a** prevents high-fat-diet-induced weight gain and improves insulin sensitivity. Our research provides strong mechanistic evidence that UCP1 is a complex mediator of **18a**-induced thermogenesis, which is a critical process in obesity mitigation. Brown adipose thermogenesis is triggered by **18a** via the AMPK-PGC-1α pathway. As a result, our research highlights a thermogenic controlled polyphenol compound **18a** and clarifies its underlying mechanisms, thus offering a potential strategy for the thermogenic targeting of adipose tissue to reduce the incidence of obesity and its related metabolic problems.

## 1. Introduction

Obesity is a global metabolic disorder with escalating prevalence, posing a severe burden to individuals and public health. Adipose tissues, which mainly consist of adipocytes, are important in controlling whole-body energy homeostasis [1,2]. Brown and beige adipocytes, which contain abundant mitochondria, play a vital role in non-shivering thermogenesis through the release of heat by uncoupling protein 1 (UCP1) [3,4]. In all mammals including humans, interscapular brown adipose tissue (iBAT) and beige adipocytes in inguinal white adipose tissue (iWAT) have the ability to perform thermogenesis and can be activated to increase energy expenditure [4]. Activating thermogenesis in these tissues constitutes an appealing target for addressing obesity and metabolic disorders [5]. UCP1, which is specifically present in brown and beige adipocytes, converts chemical energy into heat through uncoupled oxidative phosphorylation [6], thereby increasing energy expenditure and playing a key role in energy metabolism and thermoregulation. The expression and activity of UCP1 are tightly regulated by endogenous and exogenous cues [7]. These cues, which regulate the recruitment of transcription factors to the transcriptional region of UCP1 through signaling pathways, include physical stimuli, such as exposure to the cold [8] and local heat therapy [5]; endogenous factors, as well as norepinephrine (NE), 3,3′,5-Triiodo-L-thyronine (T3), and FGF21 [9]. In recent years, activation of UCP1 through small molecules of synthetic and natural origin can be considered a promising strategy for mitigating obesity [10]. These small molecules include sildenafil [11], prednisone [12], fluvastatin [13], capsaicin [14], quercetin [15], and curcumin [16].

Peroxisome-proliferator-activated receptor γ co-activator-1α (PGC-1α) is a co-transcriptional regulator that controls various aspects of energy metabolism [17], such as mitochondrial biogenesis and respiration [18]. It plays a crucial role in regulating the expression of UCP1 during the thermogenesis process in brown and beige adipocytes [19]. PGC-1α exerts direct transcriptional control over UCP1 by binding to the UCP1 gene promoter region, thereby enhancing its transcription and expression. In cellular regulation, AMPKα (AMP-activated protein kinase alpha) is an essential energy sensor and regulatory factor and has thus received considerable interest. AMPKα regulates multiple energy metabolic pathways, such as those involved in fatty acid synthesis, glucose utilization [20], and glycogen synthesis and breakdown [21], to maintain the balance in cellular energy [22]. Additionally, the activation of AMPKα is associated with that of brown adipose tissue (BAT) [23], which is important for energy expenditure and thermoregulation. AMPKα activates PGC-1α by directly binding to it and phosphorylating the Thr177 and Ser538 sites, thereby enhancing the co-transcriptional activity of PGC-1α [24].

Polyphenols, widely present in plants and food, are potential raw materials for drug development because of their high availability and low toxicity. Polyphenols exhibit various activities, including antioxidant [25], anti-inflammatory [26], anti-tumor [27], blood glucose, and lipid-regulating effects [28], among others. Additionally, numerous studies have reported that polyphenols show a prominent ability to counteract obesity [29], such as catechins [30,31] and flavonoids [32,33]. However, despite these encouraging findings, research has not yet definitively shown the direct impact of polyphenols on anti-obesity effects, particularly with regard to their role in targeting adipose tissue distribution and in directly promoting thermogenesis. This is primarily due to the inherent challenges of their high instability and limited bioavailability [34]. Cocoa polyphenols, for instance, are poorly absorbed in the intestinal tract, resulting in most of them failing to reach the body’s circulation in their original form and thus being unable to fulfill their intended functions [35]. Similarly, the bioavailability of anthocyanin is extremely low: only about 1–2% of them maintain their original structure after ingestion [36]. Resveratrol also faces the challenge of limited bioavailability, which poses a significant obstacle in translating its potential into clinical applications [37,38].

In our investigation of thermogenesis in adipose tissue, we screened polyphenolic compounds for thermogenic activity. Notably, we discovered that polyphenolic compound **18a** possesses remarkable thermogenic activity, effectively enhancing the expression of UCP1 in adipocytes. Furthermore, **18a** exhibits favorable adipose tissue distribution characteristics. To elucidate its underlying mechanisms, we performed experiments utilizing UCP1 gene knockout mice, demonstrating that the thermogenic effect of compound **18a** relies on adipose UCP1. Additionally, we investigated the involvement of the AMPK-PGC-1α signaling pathway in this thermogenesis process and its potential role in ameliorating metabolic disorders. This study offers a novel targeted strategy and an interventional approach for the treatment of obesity, thereby highlighting the significance of thermogenesis in adipose tissue.

## 2. Materials and Methods

### 2.1. Materials

The antibody sources were as follows: UCP1 (Abcam, Cambridge, UK, ab234430, 1:1000); PGC-1α (Novus, Centennial, CO, USA, NBP1-04676, 1:1000); AMPKα (Cell Signaling Technology, Danvers, MA, USA,#2532, 1:1000); phospho-AMPKα (Thr172) (Cell Signaling Technology, #2535, 1:1000); ACC (Cell Signaling Technology, #3662, 1:1000); phospho-ACC (Ser79) (Cell Signaling Technology, #3661, 1:1000); and α-Tubulin (Cell Signaling Technology, #2144, 1:5000).

DMEM/F12 Ham1:1, high glucose DMEM medium, and fetal bovine serum (FBS) were purchased from Gibco (Waltham, MA, USA). Rosiglitazone, indomycine, 3-Isobutyl-1-methylxanthine (IBMX), 3,3′,5-Triiodo-L-thyronine (T3), dexamethasone, oligomycin, carbonyl cyanide 4-(trifluoromethoxy) phenylhydrazone (FCCP), rotenone, antimycin A, cremophor EL collagenase D (Roche, Basel, Switzerland), and Dispase II (Roche) were purchased from Sigma Aldrich. Recombinant human insulin (Novolin) was purchased from Changzheng Hospital in Shanghai, China. Lipofectamine™ 3000 Transfection Reagent were from Introvigen (Grand Island, NY, USA). PrimeScript Reverse Transcriptase was purchased from Takara (Takara Ltd., Otsu, Japan). SYBR Green qPCR Master Mix was purchased from ABclonal Technology Co., Ltd. (Shanghai, China). RIPA lysis buffer, loading buffer hematoxylin and eosin were purchased from Beyotime Biotechnology Co., Ltd. (Shanghai, China).

The kits used in the measurement of plasma parameters are as follows: total triglyceride (TG) and total cholesterol (TC) were from Shanghai Fosun Long March (Shanghai, China).

### 2.2. Chemicals

18a (5-(3-(3-hexylphenyl)propyl)benzene-1,3-diol)was synthesized as previously described [39], yellow oil, ^1^H NMR (400 MHz, Chloroform-d) δ:7.20~7.16 (m, 1H), 7.03~6.96 (m, 3H), 6.28~6.22 (m, 2H), 6.18 (s, 1H), 2.62~2.50 (m, 6H), 1.95~1.87 (m, 2H), 1.64~1.56 (m, 2H), 1.34~1.28 (m, 6H), 0.88 (t, J = 6.8 Hz, 3H); ^13^C NMR (101 MHz, Chloroform-d) δ: 156.7, 145.6, 143.0, 142.1, 128.6, 128.5, 128.2, 125.8, 125.7, 108.1 (2C), 100.3, 36.0, 35.4, 35.3, 32.5, 31.7, 31.6, 29.1, 22.6, 14.1. HRMS (ESI negative) calcd for C_21_H_27_O_2_ [M-H]^−^: 311.2017, found 311.2021. Purity: 98.67% (HPLC). **18a** was synthesized by the research group led by Fa-jun Nan at the State Key Laboratory of Drug Research, the National Drug Screening Center, Shanghai Institute of Materia Medica. For the in vitro study, **18a** was dissolved in DMSO. For the in vivo study, **18a** was prepared in a solution containing 1% DMSO, 5% castor oil, and 0.5% carboxymethylcellulose sodium (CMC-Na) for oral administration to animals at a concentration of 5 mg/mL. All solutions were freshly prepared on the day of the experiment.

### 2.3. Tissue Distribution and Plasma Concentration

C57BL/6J mice were administered a dose of 20 mg/kg(p.o.). After 30 min, 20 μL of blood was collected from the retro-orbital vein of each mouse, centrifuged, and stored at −80 °C. Tissue samples from iBAT, iWAT, eWAT, the brain, heart, spleen, liver, pancreas, and kidneys were collected, rinsed, weighed, and stored at −80 °C. The samples were sent to the Servier Joint Lab for quantification of the parent compound concentration. The plasma compound levels and distribution in various tissues were measured 30 min after administration.

### 2.4. Cell Culture

C3H10-T1/2 cells provided by Ji-qiu Wang of Ruijin Hospital were cultured in high glucose DMEM supplemented with 10% FBS, with medium replacement carried out every two days. At a cell density of 80%, they were seeded into cell plates. After two days of confluence, adipocytes were induced using an induction medium consisting of basal medium supplemented with 850 nM insulin, 0.5 mM IBMX, 1 μM dexamethasone, 125 nM indomethacin, 1 nM T3, and 1 μM rosiglitazone for 2 days. Subsequently, the cells were maintained in a maintenance medium containing basal medium supplemented with 850 nM insulin, 1 nM T3, and 1 μM rosiglitazone for 6 days. On day 8, the cells were treated with **18a** for 24 h.

### 2.5. Cell Transfection

C3H10-T1/2 cells were transfected with 50 nM of siRNAs specifically targeting AMPKα or PGC-1α. The transfection process was carried out using Lipofectamine™ RNAiMAX (Invitrogen, Carlsbad, CA, USA, 13778150) following established protocols. The sequences of siRNA for PGC-1α and AMPKα were as follows: siPGC-1α-1: GUAGCGACCAAUCGGAAAUTT, AUUUCCGAUUGGUCGCUACTT; siPGC-1α-2: CCGCAAUUCUCCCUUGUAUTT, AUACAAGGGAGAAUUGCGGTT; siPGC-1α-3: CCCACAGGAUCAGAACAAATT, UUUGUUCUGAUCCUGUGGGTT; siAMPKα1: UUUGAAAGACCAAAGUCGGCU, CCGACUUUGGUCUUUCAAACA; siAMPKα2: AUCUAAACUGCGAAUCUUCUG, GAAGAUUCGCAGUUUAGAUGU.

Scramble sequences were used as the negative control.

### 2.6. Isolation of Primary Adipocytes

Primary stromal–vascular fractions (SVFs) derived from adipose tissue were isolated following the established protocol as previously described [40]. The cells were isolated from the interscapular brown fat or inguinal fat pad of 4–6-week-old mice. The collected fractions were then digested with collagenase D and Dispase II in PBS at 37 °C for 45 min. The SVFs were cultured in the DMEM/F12 medium supplemented with 10% fetal bovine serum (FBS) until reaching confluence. The differentiation medium was similar to that of C3H10-T1/2 cells, except that both the induction and maintenance media contained 10 nM T3.

### 2.7. Quantitative RT-PCR Analysis

Total RNA was isolated from cells and tissues using the Trizol reagent according to the manufacturer’s instructions. mRNA was reverse transcribed using PrimeScript Reverse Transcriptase (Nanjing Vazyme Biotech Co., Ltd., Nanjing, China, 7E782J3). The cDNA mixtures were diluted, and qPCR was performed on a Stratagene Mx3005P (Agilent, Santa Clara, CA, USA) instrument with SYBR Green qPCR Master Mix (Nanjing Vazyme Biotech Co., Ltd., Nanjing, China, 7E750G3). The qPCR protocol consisted of an initial denaturation step at 95 °C for 5 min, followed by 40 cycles of denaturation at 95 °C for 30 s, annealing at 60 °C for 30 s, and extension at 72 °C for 30 s, culminating in a final extension step at 72 °C for 5 min. Relative mRNA levels were determined using the ΔΔCT method and normalized to 36b4 gene expression. The primer sequences are provided in Appendix A.

### 2.8. Western Blotting

Cells/tissues were lysed with RIPA buffer and denatured with loading buffer. In total, 20 μg of protein was separated on 10% SDS-PAGE gel at 80 V for 30 min and then at 100 V for 60 min. Proteins were transferred to the nitrocellulose filter membrane using wet transfer at 100 V for 90 min. The membranes were blocked in TBS-T with 5% skimmed milk for 1 h, incubated with primary antibodies overnight at 4 °C and then secondary antibodies for 1 h at room temperature. Detection was carried out with an enhanced chemifluorescent substrate. Protein expression was quantified through densitometric analysis using Image J software 6.1.

### 2.9. Primary Adipocyte Respiratory Capacity Measurement

Cellular oxygen consumption was measured using the XFe 96 Extracellular Flux Analyzer (Seahorse Bioscience, Agilent, Santa Clara, CA, USA). Primary brown and beige adipocytes were differentiated until day 7, seeded into an XF 96-well microplate, and treated with a corresponding concentration of **18a** for 24 h. In total, 2 μM oligomycin, 1 μM FCCP, 1 μM antimycin, and 1 μM rotenone were used to detect uncoupled respiration, maximal respiration, and nonmitochondrial respiration, respectively.

### 2.10. Animals

The mice were housed in a temperature-controlled room (22 ± 2 °C) with a 12 h light/dark cycle and provided ad libitum access to food and water. For short-term administration experiments, thermomice were obtained from the Jackson Laboratory and randomly assigned to three groups: Vehicle, **18a** (10 mg/kg), and **18a** (20 mg/kg). The administration period lasted for seven days. In the thermoneutral experiment, the C57BL/6J mice (obtained from Beijing HFK Bioscience Co., Ltd., Beijing, Shanghai, China) were randomly divided into two groups: 22 °C and 30 °C. They were treated with either Vehicle or **18a** (20 mg/kg) for a duration of nine days. For HFD-induced obesity experiments, the male C57BL/6J mice were induced for 9 weeks (Research Diets, D12492). Subsequently, the mice were randomly divided into three groups: Vehicle, **18a** (20 mg/kg), and **18a** (40 mg/kg). The administration period lasted for nine weeks. The UCP1-KO (HO) mice [41], generously provided by Ji-qiu Wang from Ruijin Hospital, Shanghai Jiaotong University School of Medicine, along with their age-matched (WT) littermates, were treated with either Vehicle or **18a** (40 mg/kg) following a 6-week high-fat diet. The administration period lasted for ten weeks. At the end of the study, tissues were collected, weighed, and stored at −80 °C.

### 2.11. Body Composition Analysis

Body composition was assessed by using a magnetic resonance whole-body composition analyzer (Minispec LF90 II, Bruker, Karlsruhe, Germany).

### 2.12. Glucose Tolerance Test (GTT)

For the C57BL/6 high-fat diet-induced mice, GTT was performed during the 5th week. For the UCP1-KO (HO) mice, GTT was performed during the 7th week. After 6 h of fasting, the mice were treated with glucose (2.5 g/kg, p.o.), and blood glucose was recorded at different time points (15, 30, 60, 90, and 120 min) after treatment. The area under the curve (AUC) for glucose was calculated and compared between groups.

### 2.13. Insulin Tolerance Test (ITT)

For the UCP1-KO (HO) mice, the ITT was conducted during the 8th week. After fasting for 4 h, the mice were injected with insulin (0.75 U/kg, i.p.), and blood glucose levels at different time points (15, 30, 60, 90, and 120 min) after injection were recorded. The area under the curve (AUC) for glucose was calculated and compared between groups.

### 2.14. Immunohistochemical Staining

Immunohistochemical staining was performed on formalin-fixed paraffin-embedded sections. For antigen recovery, the sections were incubated with ethylenediaminetetraacetic acid (EDTA, PH 9.0) for 5 min at 120 °C, followed by 3% H_2_O_2_-methanol for 10 min at room temperature. The sections were closed with goat serum for 30 min. The sections were incubated with anti-UCP1 (ab234430, brown adipose tissue 1:6400, beige adipose tissue 1:400; Abcam, Cambridge, UK) wet cassettes at 4 °C overnight. The secondary antibody working solution (111-035-003, Jackson, MS, USA, Peroxidase AffiniPure Goat Anti-Rabbit IgG (H+L)) was incubated in a wet box at room temperature for 30 min, color-developed by DBA (Sigma-Aldrich, St. Louis, MO, USA, D8001) for 5–10 min, washed with water for 15 min, and then restained with hematoxylin. This process also involved gradient alcohol dehydration, xylene transparency, and neutral gum sealing.

### 2.15. Histology

The adipose tissues were fixed in 4% formaldehyde at room temperature overnight. After fixation, the tissues were embedded in paraffin and sectioned into slices with a thickness of 6 μm using a microtome. The sections were then subjected to hematoxylin and eosin staining following the manufacturer’s instructions. Images of the sections were captured using a Leica DM6 B microscope. The cell surface area was calculated using the AdipoCount application reported before [42] http://www.csbio.sjtu.edu.cn/bioinf/AdipoCount/.

### 2.16. Statistical Analysis

Data are presented as the mean ± SEM. The two-tail Student’s *t*-test was used for comparisons between two groups and one-way ANOVA was used for comparisons of more than two groups, followed by Tukey’s multiple comparison test. Statistical significance was considered when *p*-values < 0.05. Analyses and figures were produced using GraphPad Prism software (version 7.00, GraphPad Software, La Jolla, CA, USA).

## 3. Results

### 3.1. ***18a*** Induces the Thermogenic Program in C3H10-T1/2 Cells While Exhibiting Favorable Adipose Tissue Distribution

Polyphenols, including resveratrol, modulate mitochondrial processes to promote thermogenesis and enhance energy expenditure, potentially aiding in obesity management [43,44]. Consequently, we conducted a screening of polyphenolic compounds for UCP1 transcription levels, which resulted in the identification of a promising thermogenic compound, **18a** (Appendix A). Furthermore, we found that **18a** exhibits intense plasma exposure (Figure 1A) and favorable adipose tissue distribution characteristics, including iWAT, iBAT, and epididymal adipose tissue (eWAT) (Figure 1B). When treating the C3H10-T1/2 cells for 24 h, we found that **18a** (Figure 1C) significantly increased the transcriptional level of UCP1 and thermogenic factors, including *Pgc-1α*, peroxisome proliferator-activated receptor α (*Pparα*), PRD1-BF-1-RIZ1 homology structural domain protein 16 (*Prdm16*), type II thyroxine deiodinase (*Dio2*), cytochrome C oxidase subunit 7a1 (*Cox7a1*), cytochrome C oxidase subunit 8b (*Cox8b*), cell death-inducing DNA fragmentation factor-like effector a (*Cidea*), and estrogen-related receptor α (*Errα*) (Figure 1D) at non-toxic doses (Appendix A). Furthermore, **18a** treatment led to a significant increase in UCP1 protein levels, along with notable upregulation of PGC-1α (Figure 1E–G). In summary, these findings collectively indicate that **18a** stimulates the thermogenesis program in C3H10-T1/2 cells and demonstrates advantages in tissue distribution that are not found in other polyphenols.

### 3.2. ***18a*** Enhances Thermogenesis and Mitochondrial Respiration in Primary Brown and Beige Adipocytes

To validate the thermogenic activation potential of **18a**, we isolated the stromal vascular fraction (SVF) from iBAT, differentiated it into mature brown adipocytes, and then exposed it to **18a**. This led to an increase in the thermogenic genes *Ucp1*, *Pgc-1α*, *Pparα*, *Prdm16*, *Cox7a1*, *Cox8b*, *Cidea*, and *Errα* (Figure 2A). Additionally, **18a** upregulated the protein levels of UCP1 and PGC-1α (Figure 2B). Subsequently, the Seahorse XF analysis system was employed to measure the cellular oxygen consumption rate (OCR) and to investigate the enhanced energy expenditure effect of **18a**. The results revealed that primary brown adipocytes treated with **18a** exhibited significantly higher basal and uncoupled respiration rates compared to those of untreated cells (Figure 2E,F). In primary beige adipocytes, the function of **18a** (Figure 2G–L) was similar to that observed in primary brown adipocytes. Collectively, these findings indicate that **18a** enhances the thermogenic program and mitochondrial respiratory capacity of both brown and beige adipocytes.

### 3.3. Short-Term ***18a*** Treatment Significantly Activates Thermogenesis and Reduces Adipose Tissue Weight in Mice

In vitro, the thermogenic function of **18a** was confirmed in primary brown and beige adipocytes; moreover, we used thermomice to investigate **18a**’s function in vivo. The thermomice, carrying the *Ucp1* gene sequence fused with the luciferase gene sequence, represent an in vivo model that identifies the modulators of UCP1 expression in brown adipose tissue [45]. After one week of **18a** treatment, the mice showed a significant increase in fluorescence intensity in their interscapular region (Figure 3A–C), as well as higher rectal and iBAT skin temperatures during cold exposure (Figure 3D–F). Adipose tissue from **18a** mice showed both a significant reduction (Figure 3G) and increased density in all adipose tissue when analyzed through histological staining (Figure 3H–J). Additionally, iBAT and iWAT after the administration of **18a** revealed a significant increase in UCP1 expression levels through immunohistochemical staining (Figure 3K), as well as increased levels of thermogenic gene expression both in mRNA (Figure 3L,M) and protein levels (Figure 3N,O). In summary, **18a** administration substantially augmented the expression of UCP1 in the adipose tissue of mice and spurred thermogenesis, thereby resisting body fat accumulation and inducing whole-body energy expenditure.

### 3.4. Thermogenic Effects Induced by ***18a*** Treatment Are Ablated in Thermoneutral Conditions

The previous results demonstrate that the administration of **18a** significantly enhanced the expression of UCP1 both in vivo and in vitro. The objective is to investigate whether the compound’s ability to reduce body fat reserves in mice operates via a thermogenic pathway. In thermoneutral conditions, BAT becomes “inactive” to protect against cold stimuli [3]. Indeed, a thermoneutral comparison study was conducted on mice kept within an environment of either 22 °C or 30 °C following the administration of a high-fat diet (Figure 4A). After one week of treatment, the 22 °C group revealed their ability to maintain core and interscapular skin temperatures when compared with the 30 °C group. In fact, mice treated with **18a** in 30 °C conditions displayed a diminished ability to maintain core body and interscapular skin temperatures during cold exposure compared with the group treated at 22 °C (Figure 4B–D). This study further analyzed fat content, and mice in 30 °C conditions revealed a significant reduction in the weight of their iBAT, with higher iWAT and eWAT than the 22 °C group. In comparison with the Vehicle group, a reduction in the weight of adipose tissue was achieved by **18a** at 22 °C, yet this was no longer evident at 30 °C (Figure 4E). This observation was additionally supported through the analysis of body fat composition (Figure 4F). Taken together, these findings indicate that **18a**’s effect relies on heat production by thermogenic adipocytes.

### 3.5. ***18a*** Resists High-Fat-Induced Weight Gain and Enhances Glucose Tolerance in Mice

Based on the aforementioned effect in vitro and in vivo, C57BL/6J mice were nourished with high-fat food to investigate the therapeutic effect of **18a** on diet-induced obesity. Following 5-week treatment with a dose of 40 mg/kg per day, the mice had significantly reduced body weights (Figure 5A) and a significant reduction in fat content (Figure 5B) with no difference in food intake (Appendix A). Metabolic syndrome, which is characterized by insulin resistance, hyperglycemia, and hyperlipidemia, is commonly found alongside obesity and leads to significant tissue damage [46]. Weight loss often coincides with improvements in insulin resistance and hyperlipidemia. A high-fat diet resulted in decreased glucose tolerance, whereas **18a** substantially ameliorated abnormal glucose metabolism (Figure 5C,D). Furthermore, **18a** treatment significantly reduced plasma concentrations of total cholesterol (TC) and triglyceride (TG) (Appendix A). The abovementioned results indicate that **18a** prevents high-fat-diet-induced obesity and lessens the effects of systemic glycemic and lipemic disorders. Adaptive thermogenesis plays a crucial role in maintaining body temperature during cold stimulation through uncoupled fuels burning with ATP synthetics, thereby being considered to be a potential anti-obesity strategy. Notably, mice treated with 18a demonstrated higher rectal and iBAT skin temperatures during cold exposure (Figure 5E–G). Meanwhile, **18a** significantly reduced adipose tissue weight (Figure 5H) and adipocyte areas (Figure 5I–K). In conclusion, **18a** can enhance energy expenditure by activating thermogenesis in adipose tissue, resist high-fat diet-induced obesity, and lessen the effects of glycolipid metabolic disorders.

### 3.6. Deficiency in UCP1 in Mice Reverses the Metabolic Benefits of ***18a***

As previously mentioned, **18a** significantly upregulates adipose UCP1 and mitigates body fat gain dependent on thermogenesis. UCP1 is a key thermogenic protein, but adipose UCP1-independent thermogenesis, such as SERCA2b-mediated calcium cycling [47] and creatine-dependent substrate cycling [48], also play an important role in thermogenic fat. To investigate the mode of **18a** promoting adipose thermogenesis, we used UCP1 knockout mice (HO) to examine the function of **18a**. Consistent with prior studies [49], HO mice exhibited a significantly poor ability to deal with the cold but also a lower rate of weight gain and superior insulin sensitivity (Figure 6E,F), which were potentially associated with the higher levels of adiponectin and FGF21 sensitivity observed in HO mice [50]. This study revealed that HO mice did not experience all of the metabolic benefits observed in response to **18a** treatment, such as weight loss (Figure 6A), reduced body fat content (Figure 6B), improved glucose metabolism (Figure 6C–F), and enhanced cold tolerance (Figure 6G–I). Subsequent isolation of adipose tissue demonstrated that **18a** administration did not decrease adipose tissue weight (Figure 6J) in HO mice, as indicated by the H&E results (Figure 6K) and cell area statistics (Figure 6L,M). These findings indicate that the presence of UCP1 is crucial for **18a**’s capacity to promote thermogenesis, mitigate body fat gain, and enhance glycolipid metabolism.

### 3.7. ***18a*** Exerts Thermogenesis-Promoting Effects through AMPK-PGC-1α

Multiple studies have reported that polyphenols, targeting AMPK signaling, play an important role in the treatment of obesity [51]. Our results demonstrated that **18a** dose-dependently enhances the phosphorylation level of AMPKα and its downstream ACC (Figure 7A), without significant effects on other signaling pathways involved in thermogenesis regulation (Appendix A). Moreover, the AMPK inhibitor dorsomorphin effectively blocked the expression of UCP1 and PGC-1α induced by **18a** (Figure 7B). To further validate the involvement of AMPKα in the thermogenic effect of **18a**, we employed small interfering RNA to knock down AMPKα in C3H10-T1/2 cells before **18a**. The results revealed that AMPKα depletion suppressed the **18a**-induced expression of UCP1 and PGC-1α at both mRNA (Figure 7C) and protein levels (Figure 7D–F). AMPK regulates adipose thermogenesis through PGC-1α post-translational modifications and the subsequent regulation of positive feedback transcriptional loops [24,52]. By using small interfering RNA to silence PGC-1α in C3H10-T1/2 cells, we found that silencing PGC-1α impacts the promotion of UCP1 expression by **18a**, at both the mRNA (Figure 7G) and protein level (Figure 7H). These findings indicate that **18a** activates adipose thermogenesis via AMPK-PGC-1α.

In summary, we have elucidated how the polyphenol **18a** enhances the adipose thermogenic program through the AMPK-PGC-1α pathway, and these effects are UCP1 dependent. Furthermore, we demonstrated that the beneficial effects of **18a** in combating obesity and improving insulin resistance are dependent on the thermogenic processes occurring in adipose tissues. Collectively, these findings support the further development of polyphenols as a potential therapeutic for obesity and metabolic diseases.

## 4. Discussion

Polyphenols have received attention for their diverse activities, particularly their potential anti-obesity effects. Notably, Jaboticaba phenolic extract has demonstrated protective effects against insulin resistance and dyslipidemia, preventing obesity induced by a high-fat diet [53]. Additionally, research has shown that resveratrol significantly upregulates UCP1 protein expression in adipose tissue, activating non-shivering thermogenesis in brown adipocytes and, in turn, reducing body weight [54]. Similar benefits were observed for daidzein treatment, which reduced weight by increasing UCP1 expression [55]. Our research findings also indicate that the polyphenolic compound **18a** can increase UCP1 expression in adipose tissue, thus promoting thermogenesis, increasing energy expenditure, and reducing adipose tissue content. Furthermore, these effects of **18a** in combating obesity are dependent on UCP1-dependent thermogenesis. Accordingly, there is a compelling likelihood that leveraging polyphenols to boost UCP1-dependent thermogenesis could play a significant role in combating obesity.

Despite the therapeutic potential of polyphenols, their limited bioavailability poses a challenge to their clinical use [56,57,58,59]. Polyphenols face obstacles like water insolubility, limited absorption, and rapid metabolism, leading to low oral bioavailability [60]. To overcome this, various drug delivery strategies may enhance bioavailability and boost the therapeutic effectiveness of polyphenolic compounds. These approaches include pro-drug designs [61], formulations with cyclodextrin [62], simple emulsions, self-emulsifying delivery systems, gels, lipid nanocapsules [63], nanoemulsions [64], and liposomes [65,66]. Furthermore, the discovery of polyphenolic compounds with improved bioavailability is also crucial. For instance, compound **18a** exhibits excellent bioavailability and targeted distribution to adipose tissue, making it a promising candidate for combating obesity.

Numerous studies have reported that polyphenols activate AMPK’s phosphorylation activity [51,67] and have potential applications in aging [68] and cancer prevention [69]. Up-regulating AMPK in adipocytes, as an energy regulator, facilitates thermogenesis in brown and beige fat and improves diet-induced glucose intolerance [70]. In addition, the phytochemical hyperforin triggers thermogenesis in adipose tissue via a Dlat-AMPK signaling axis to curb obesity [51]. Similarly, tea polyphenols exhibit beneficial effects in reducing adiposity and inflammation and enhancing insulin sensitivity. These effects are closely associated with their stimulatory effects on the FGF21/AMPK/UCP1 pathway [71]. Our research has revealed that **18a** promotes UCP1-dependent thermogenesis through the AMPK-PGC-1α pathway, enhancing resistance to obesity. Based on these discoveries, we propose that polyphenols can also be employed as a targeted AMPK strategy against obesity by enhancing thermogenic processes.

Research on the mechanisms underlying non-shivering thermogenesis (NST) has gained momentum due to the growing interest in metabolic disorders associated with obesity, such as type 2 diabetes and fatty liver disease [72]. Adaptive thermogenesis comprises the generation of heat by the body in response to external stimuli, and it offers a potential strategy to counteract the hypercaloric state of obesity. Adaptive thermogenesis can be categorized into shivering and non-shivering forms, with the latter being of particular interest in this context. UCP1 is the most prominent effector of non-shivering thermogenesis and plays a vital role in this process [7]. In our investigations, administration of 18a consistently resulted in a significant increase in endogenous UCP1 levels and enhanced the transcriptional activity of the UCP1 promoter in vitro (Appendix A). Using HO mice, we observed that **18a** led to a substantial reduction in the rate of body weight gain and body fat content in WT mice, but this was not observed in HO mice, indicating that the positive effects of **18a** on whole-body glucose and energy homeostasis are mediated through UCP1. Moreover, we unexpectedly discovered that the rate of body weight gain in UCP1 KO mice was much lower than that in WT mice, which aligns with previous findings [49]. This finding is associated with the obesity resistance of UCP1-deficient mice, which is characterized by persistent FGF21 sensitivity in inguinal adipose tissue [73]. Additionally, one study demonstrated that AMPK activation induces UCP1-independent thermogenesis in subcutaneous white adipose tissue, effectively preventing diet-induced obesity [74]. This effect was shown to be mediated through the creatine cycle and the Ca2^+^ cycle. However, there is a debate regarding whether **18a** induces UCP1-independent thermogenesis via AMPK. Our experimental findings demonstrate the lack of upregulation in genes associated with UCP1-independent thermogenesis upon **18a** administration. This result (Appendix A) aligns with the ineffectiveness of **18a** in reducing body fat in HO mice. These results suggest that **18a** may not exert its thermogenic effects through UCP1-independent pathways, further emphasizing the importance of UCP1 in mediating the metabolic benefits observed with **18a** treatment.

Our studies have demonstrated that the polyphenolic compound **18a** exerts its effects on thermogenesis by modulating UCP1-dependent programs in iBAT and iWAT. This highlights the significance of UCP1 as a key mediator of thermogenesis in the context of combating obesity with **18a**. However, the limitations of our study must be acknowledged. One such limitation is the use of UCP1 KO mice (HO), as this approach abolishes the primary thermogenic mechanism in these animals and may trigger compensatory regulatory mechanisms for thermogenesis. Indeed, the reduction in respiratory chain abundance is not confined to the brown adipose tissue [75] but is also observed in beige adipose tissue [7,76] in UCP1-KO mice. It is important to recognize that the constitutive UCP1-KO mouse model may not be ideal for specifically studying UCP1 function in vivo but rather serves as a broader tool for investigating global dysfunction in both brown and beige adipose tissue [7]. To fully elucidate the underlying mechanisms at play in our UCP1-KO mice model, further investigations are imperative.

## 5. Conclusions

In conclusion, our research presents the first evidence that the polyphenol compound **18a** improves thermogenesis in adipose tissue by modulating UCP1-dependent programs, leading to beneficial effects on weight loss and energy expenditure through the adipose AMPK-PGC-1α signaling pathway. Furthermore, it demonstrates excellent bioavailability, enabling the efficient targeting of adipose tissue. These findings not only deepen our comprehension of polyphenol biological activities but also introduce a promising small-molecule compound with potential anti-obesity effects.

## Figures and Tables

**Figure 1 biomolecules-14-00618-f001:**
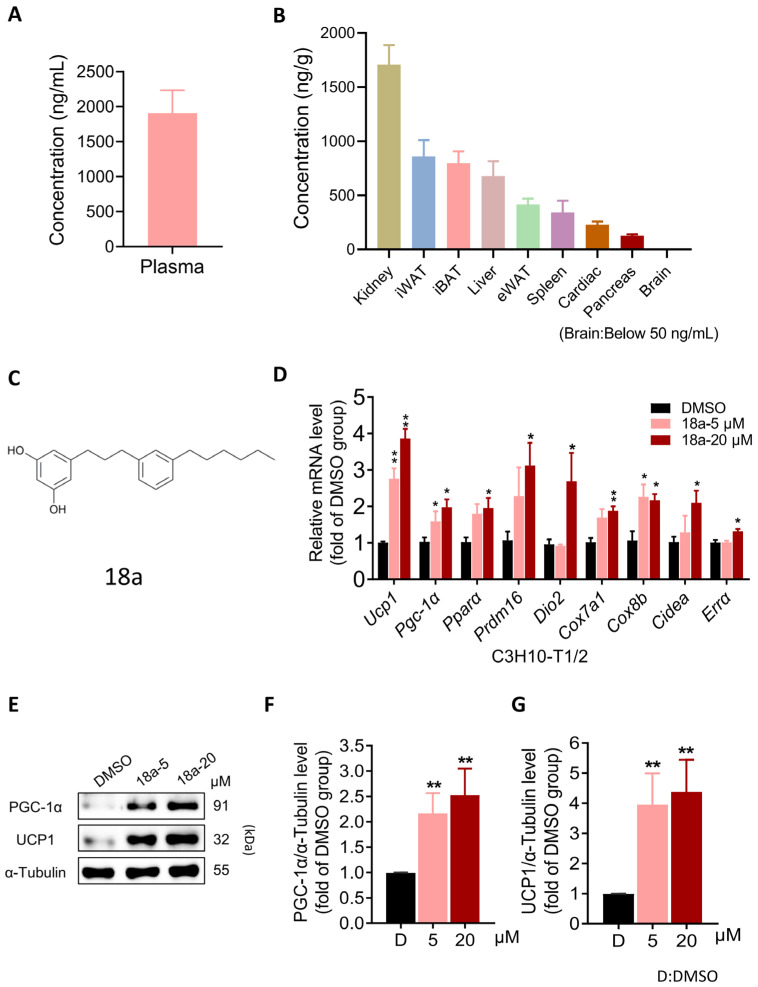
**18a** exhibits favorable distribution in adipose tissues and activates UCP1 expression in C3H10-T1/2. (**A**) The plasma concentration of compound **18a** in C57BL/6J mice at 30 min following oral administration of 20 mg/kg; (**B**) the tissue distribution of **18a** in C57BL/6J mice; (**C**) structure of **18a**; (**D**) effect of **18a** on the transcriptional levels of thermogenic genes; (**E**) expressions of indicated protein by **18a**; (**F**,**G**) protein level statistics of PGC-1α (**F**), UCP1 (**G**) compared to α-tubulin. *n* = 4 per group. Data presented as the means ± SEM. * *p* < 0.05 and ** *p* < 0.01 versus the DMSO group. Original images can be found in Appendix A.

**Figure 2 biomolecules-14-00618-f002:**
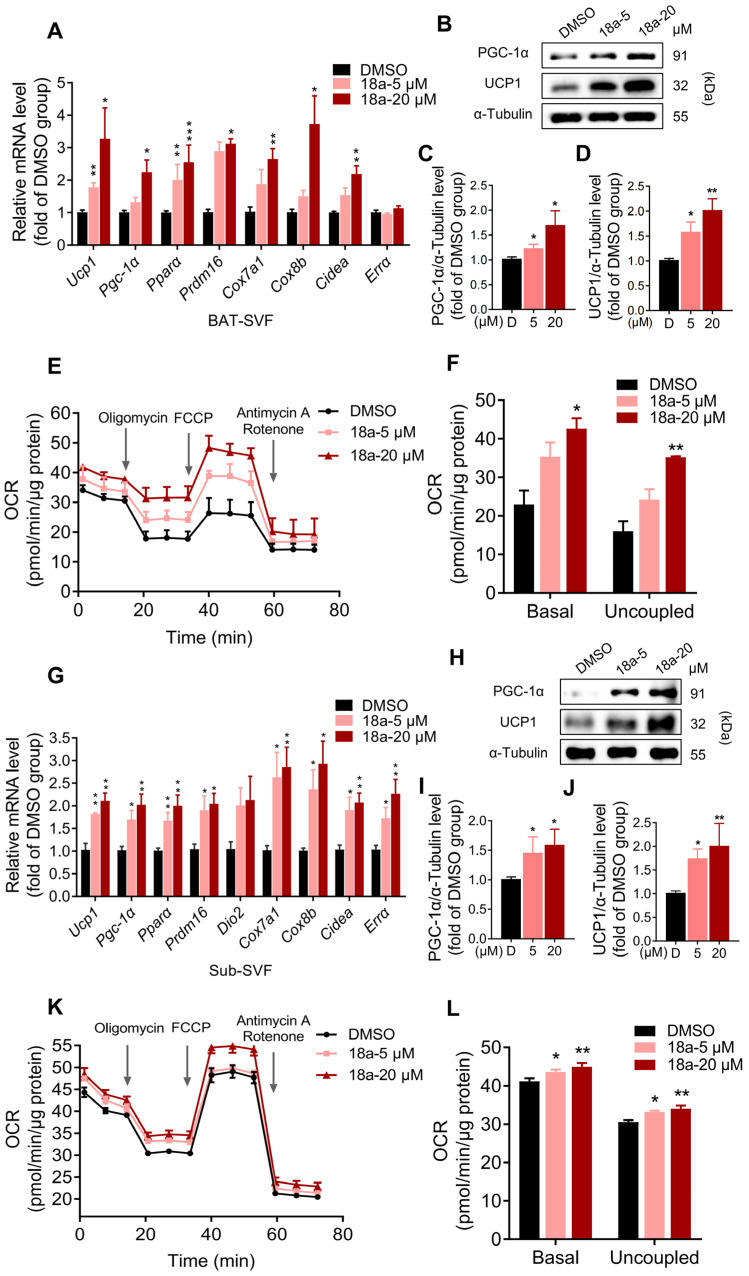
**18a** enhances the thermogenic program and increases mitochondrial respiration on primary brown and beige adipocytes. Effect of **18a** on (**A**) mRNA levels of thermogenic genes in primary brown adipocytes; (**B**) protein levels of PGC-1α and UCP1 expression in primary brown adipocytes; (**C**,**D**) comparison of relative protein levels of PGC-1α (**C**) and UCP1 (**D**) compared to α-tubulin; (**E**) detection of OCR; (**F**) basal respiration and uncoupled respiration statistics; (**G**) mRNA levels of thermogenic genes in primary beige adipocytes; (**H**) protein levels in primary beige adipocytes; (**I**,**J**) comparison of relative protein levels of PGC-1α (**I**) and UCP1 (**J**) compared to α-tubulin; (**K**) detection of OCR; (**L**) basal respiration and uncoupled respiration statistics; *n* = 4 per group. * *p* < 0.05, ** *p* < 0.01, and *** *p* < 0.001 versus the DMSO group. Original images can be found in Appendix A.

**Figure 3 biomolecules-14-00618-f003:**
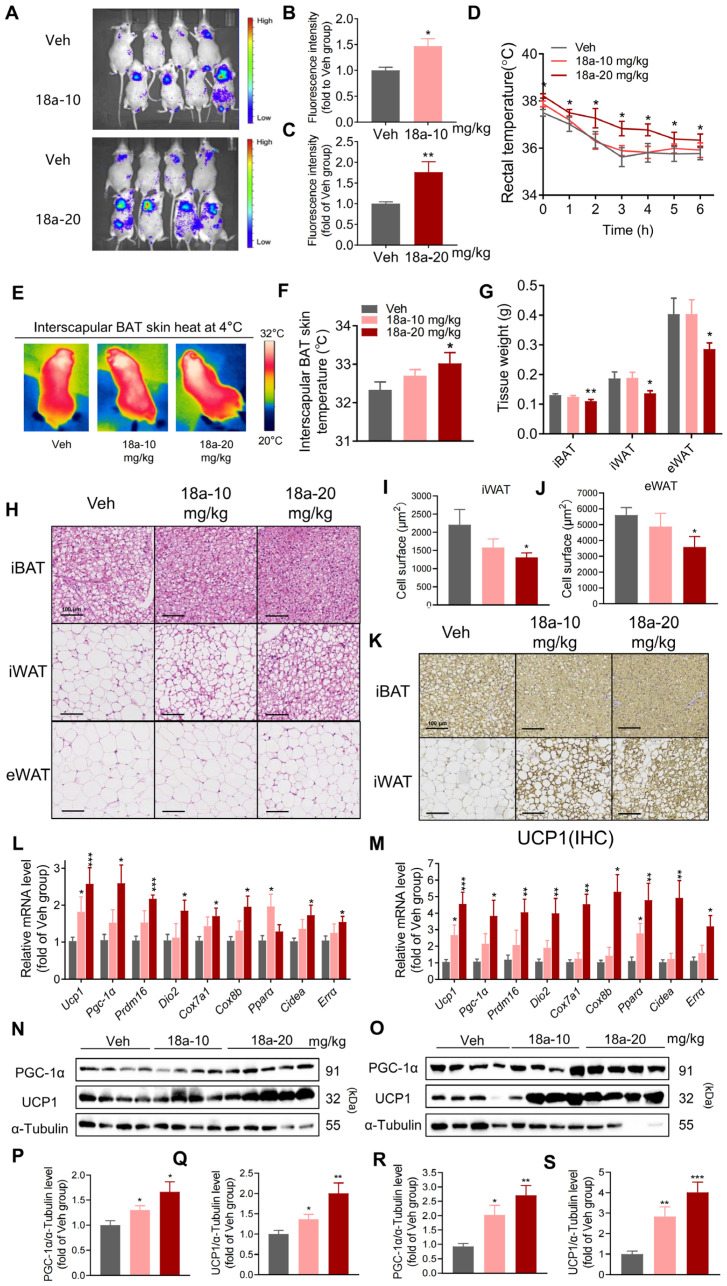
**18a** enhances the activation of the thermogenic program and reduces adipose tissue weight in mice. (**A**) UCP1 reporter gene expression after **18a** administration; (**B**,**C**) fluorescence statistics; (**D**) rectal temperature changes in mice after 4 °C exposure; (**E**) infrared thermal images of mice after 6 h cold stimuli; (**F**) quantitative statistics of interscapular skin surface temperature; (**G**) weights of iBAT, iWAT, and eWAT in mice; (**H**) H&E staining of adipose tissues; (**I**) cell area counts of iWAT; (**J**) Cell area counts of eWAT; (**K**) immunohistochemical staining of UCP1 in iBAT and iWAT; (**L**) detection of thermogenic mRNA levels in iBAT; (**M**) detection of thermogenic mRNA levels in iWAT; (**N**,**O**) Western blot analysis of indicated proteins in iBAT (**N**) and iWAT (**O**); (**P**,**Q**) comparison of PGC-1α (**P**) and UCP1 (**Q**) protein levels compared to α-tubulin in iBAT (**R**,**S**); comparison of PGC-1α (**R**) and UCP1 (**S**) protein levels compared to α-tubulin in iWAT. *n* = 4–6 per group, * *p* < 0.05, ** *p* < 0.01, and *** *p* < 0.001 versus the Vehicle group. Original images can be found in Appendix A.

**Figure 4 biomolecules-14-00618-f004:**
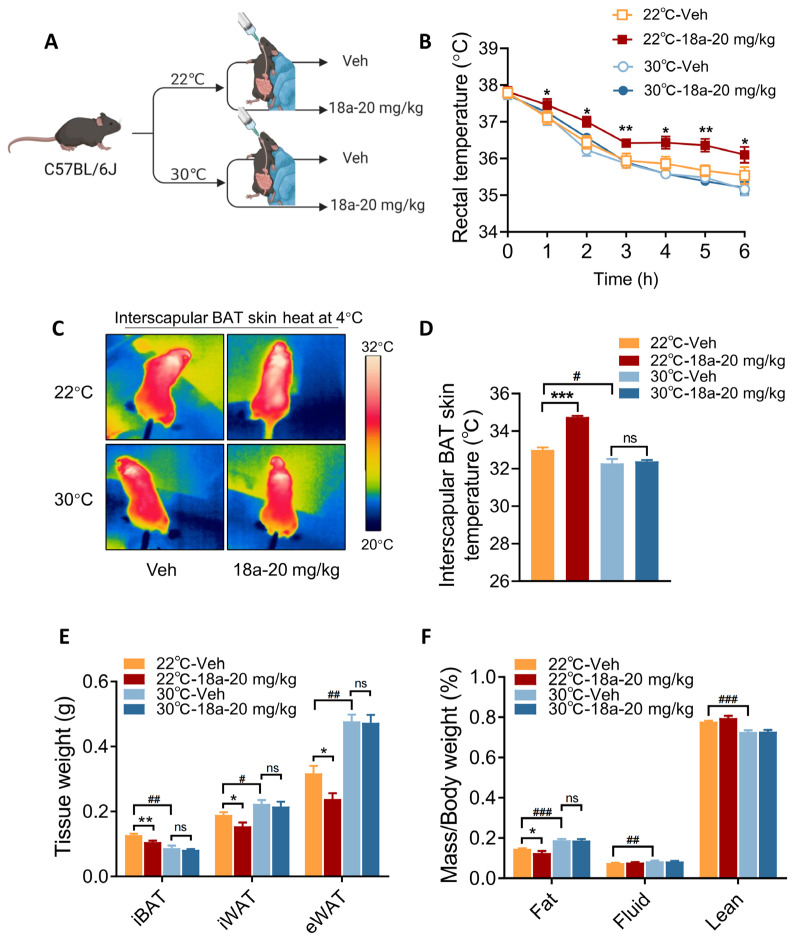
The thermogenic effects of **18a** are nullified under thermoneutral conditions. (**A**) Schematic diagram of the experimental procedure; (**B**) rectal temperature of 22 °C and 30 °C mice subjected to cold stimuli; (**C**) representative pictures of infrared thermal images; (**D**) statistics of the interscapular skin temperatures of the mice; (**E**) weights of iBAT, iWAT, and eWAT; (**F**) percentage of body composition as a percentage of body weight; *n* = 6, * *p* < 0.05, ** *p* < 0.01, *** *p* < 0.001, ns = non-significant. **18a** group versus the Vehicle group; # *p* < 0.05, ## *p* < 0.01, and ### *p* < 0.001, 30 °C group versus the 22 °C group.

**Figure 5 biomolecules-14-00618-f005:**
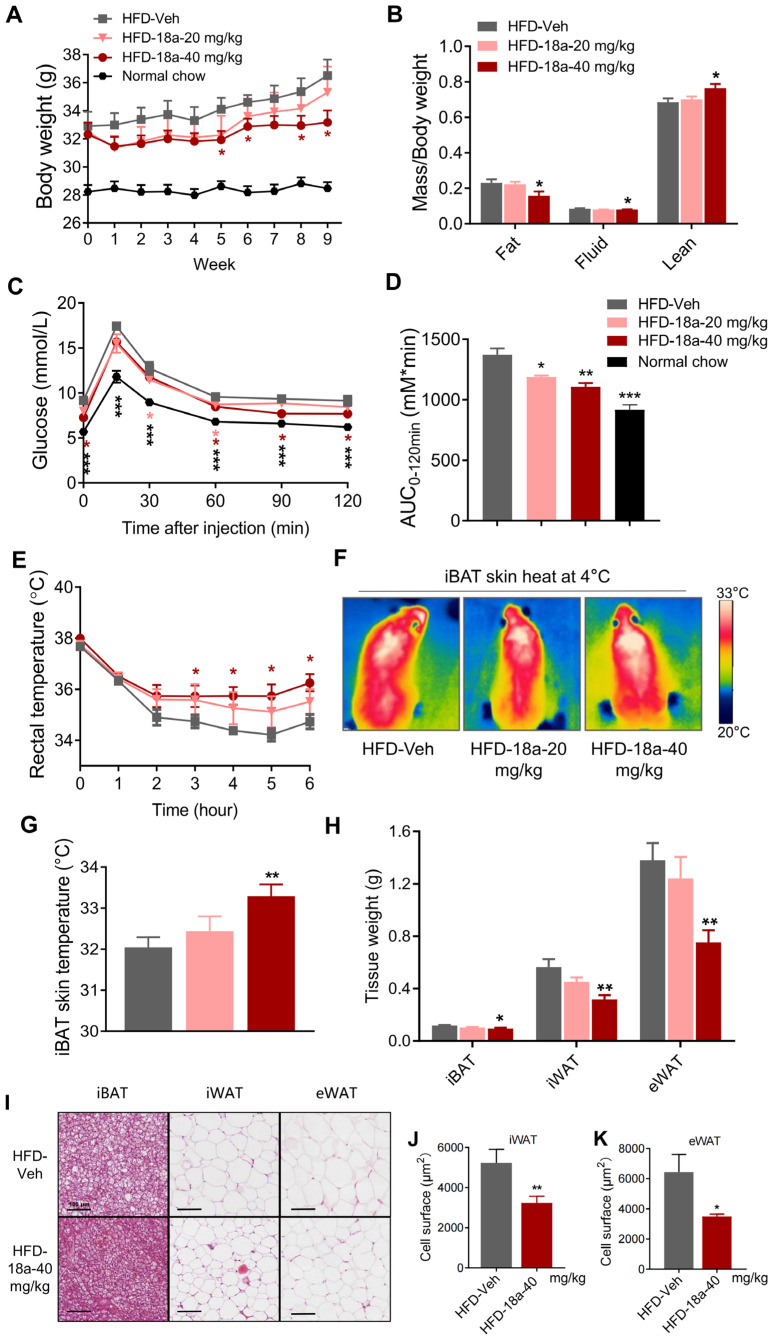
**18a** resists weight gain and enhances glucose tolerance in obese mice. (**A**) Body weight changes per week; (**B**) body mass index statistics; (**C**) oral glucose tolerance test (GTT); (**D**) area under the curve (AUC) of GTT; (**E**) changes in rectal temperature after 4 °C exposure; (**F**) infrared thermal images in the HFD-Vehicle or HFD-**18a** group after exposure to 4 °C condition for 6 h; (**G**) quantitative statistics of interscapular skin surface temperature; (**H**) weights of adipose tissues; (**I**) representative hematoxylin and eosin staining from iBAT, iWAT, and eWAT sections; (**J**,**K**) quantification of adipocyte area of iWAT (**J**) and eWAT (**K**). *n* = 6, * *p* < 0.05, ** *p* < 0.01 and *** *p* < 0.001 versus HFD-Vehicle group.

**Figure 6 biomolecules-14-00618-f006:**
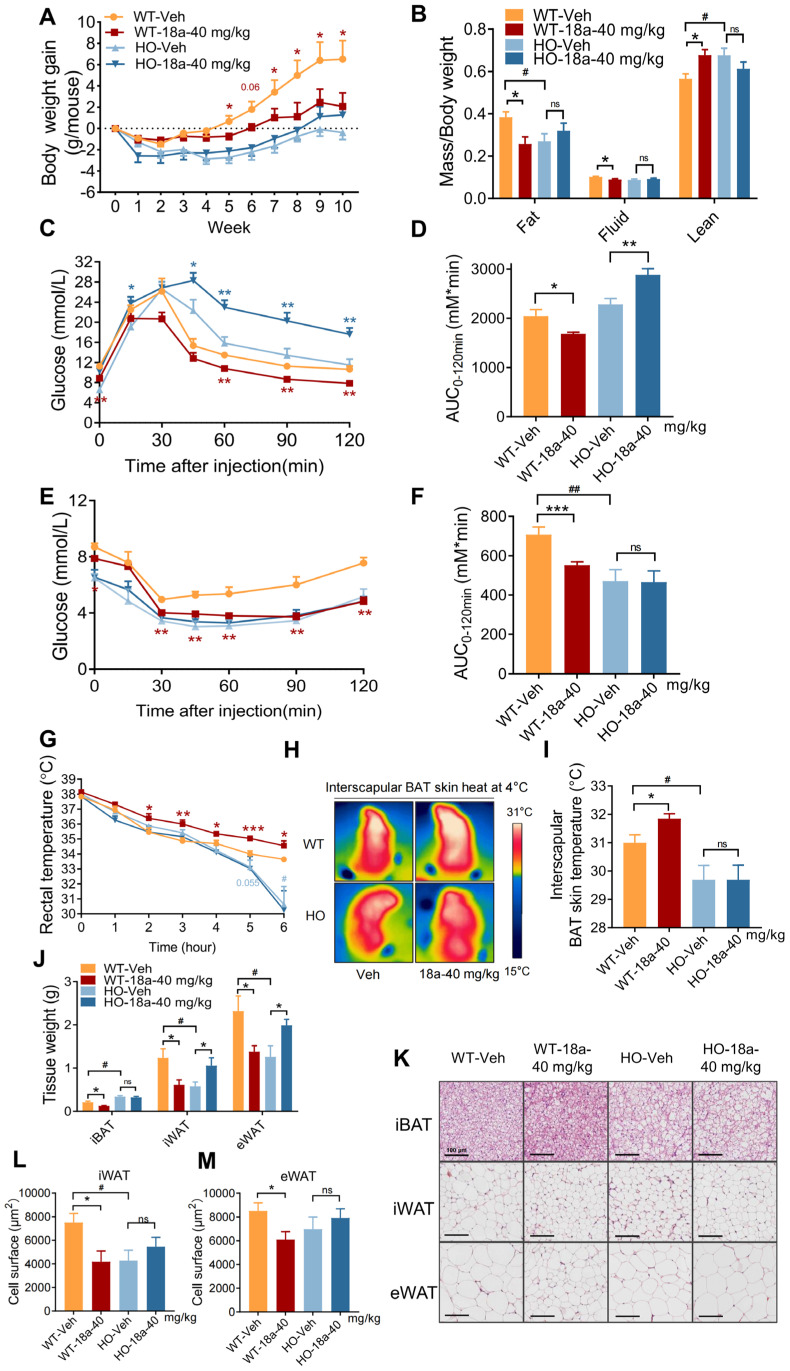
UCP1 deficiency reverses the metabolic benefits of **18a**. (**A**) Body weight of high-fat diet WT and HO mice after **18a** treatment; (**B**) body mass index statistics; (**C**,**D**) GTT (**C**) and AUC statistics of WT and HO mice (**D**); (**E**,**F**) ITT (**E**) and AUC statistics of WT and HO mice; (**F**); (**G**) rectal temperature changes of WT and HO mice upon exposure to cold stimuli; (**H**) representative images of infrared thermography of mice exposed to 4 °C condition for 6 h; (**I**) iBAT skin temperature statistics; (**J**) adipose tissue weight of WT mice and HO mice; (**K**) H&E staining of adipose tissue; (**L**) iWAT cell area statistics; (**M**) eWAT cell area statistics; *n* = 5–6, * *p* < 0.05, ** *p* < 0.01, *** *p* < 0.001 and ns = non-significant, **18a** group versus the Vehicle group. # *p* < 0.05 and ## *p* < 0.01, WT-Vehicle group versus the HO-Vehicle group.

**Figure 7 biomolecules-14-00618-f007:**
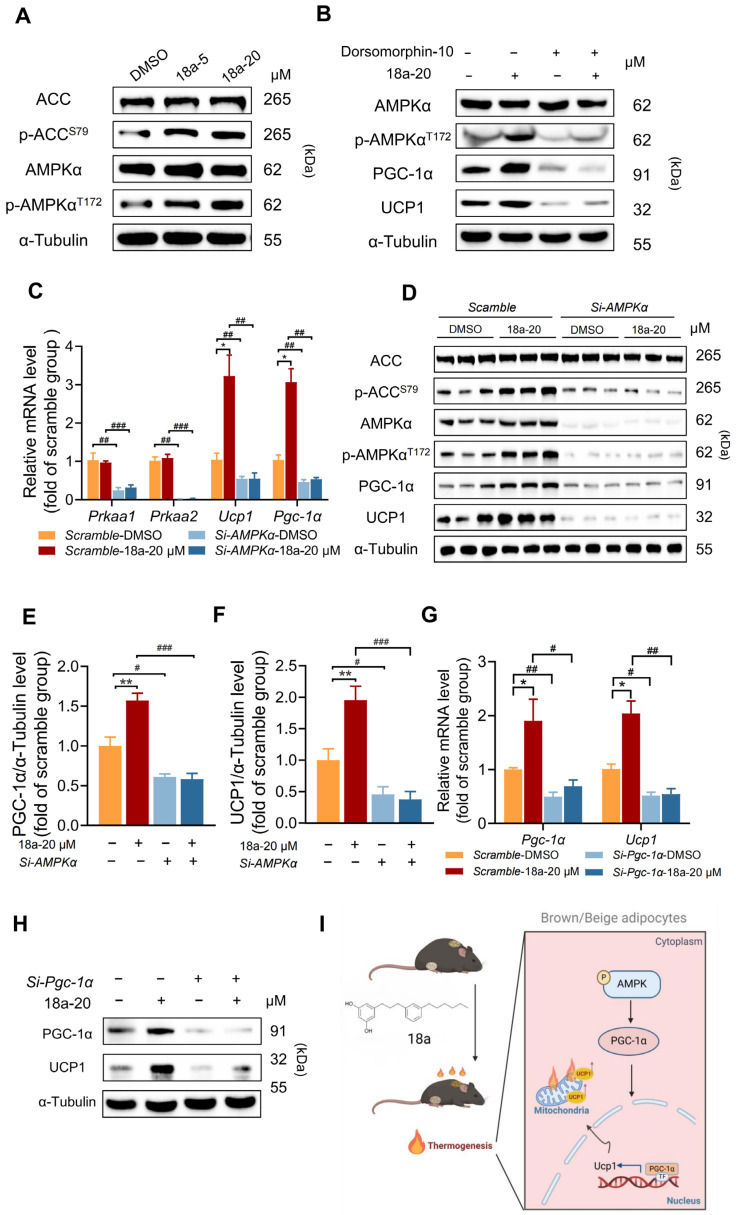
**18a** exerts thermogenesis-promoting effects through AMPK-PGC-1α. (**A**) Western blot of p-AMPKα, AMPKα, p-ACC, and ACC in **18a**-treated C3H10-T1/2; (**B**) target proteins in C3H10-T1/2 treated with dorsomorphin followed **18a**; (**C**) mRNA levels of Pgc-1α and Ucp1 after AMPKα knockdown; (**D**) expression of PGC-1α and UCP1 proteins in C3H10-T1/2 after AMPKα knockdown; (**E**,**F**) comparison of PGC-1α (**E**) and UCP1 (**F**) protein levels to α-Tubulin; (**G**) mRNA levels of Pgc-1α and Ucp1 after PGC-1α knockdown; (**H**) protein levels of PGC-1α and UCP1 after PGC-1α knockdown; (**I**) graphical abstract. *n* = 3 per group. Data are presented as the means ± SEM. * *p* < 0.05, ** *p* < 0.01, **18a** group versus the DMSO group; # *p* < 0.05, ## *p* < 0.01, and ### *p* < 0.001, siRNA group versus the Scramble group. Original images can be found in Appendix A.

## Data Availability

The data are contained within the article.

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
