# Peer review of "Polyphenol Compound 18a Modulates UCP1-Dependent Thermogenesis to Counteract Obesity"

_biomolecules, 2024, doi:10.3390/biom14060618_

Round 1
Reviewer 1 Report
Comments and Suggestions for Authors
This article studied a chemical called 18a in adipocytes.
Brown and beige adipose tissues have been studied for their ability to increase energy expenditure through thermogenesis, primarily mediated by the expression of uncoupling protein 1 (UCP1). Stimulating these tissues could offer a novel approach to treating metabolic diseases like obesity.
The study identifies a specific polyphenol compound called 18a, which has been shown to activate UCP1, thereby promoting thermogenesis and enhancing mitochondrial respiration in brown adipocytes. The authors also provided mechanistic insights into how compound 18a induces thermogenesis, particularly through the AMPK-PGC-1α pathway, highlighting the molecular pathways involved in its beneficial effects on adipose tissue function.
Not only the vivo study in animal studies, compound 18a is demonstrated to prevent weight gain induced by a high-fat diet and improve insulin sensitivity, suggesting its potential therapeutic efficacy in combating obesity and associated metabolic disorders.
The experiments in this study were well-designed, and the results obtained were clear, providing strong support for the authors' hypothesis and leading to a clear and definitive conclusion.
The study will be appreciated for developing new medication or management against obesity, and I recommend publishing this manuscript in Biomolecules after these minor revisions.
Given the key role of 18a in the study, it is crucial to provide comprehensive data on its synthesis and identification. Specifically, the compound identification data, which confirms the structure and purity of 18a, should be included. This will enhance the credibility of the study and its findings.
The title said, novel polyphenol compound 18a. I understand the authors numbered the compounds when they tested, and 18a was the best. However, as a publication, the compound number should start with 1.
Show the structure of 18a when it is mentioned, not in the supporting information. The structure is shown in Figure 7, and it is close to the end of the manuscript. Because "18a" does not indicate any detailed structural information, it is better to introduce the structure at the beginning so that the reader can easily compare it with other polyphenol structures.
The discussion contains a significant amount of repetitive content from the introduction. To enhance the clarity and flow of your manuscript, consider removing the detailed research background from the discussion section. This will help to avoid repetition and ensure a more streamlined presentation of your findings.
The first paragraph states a similar thing at the beginning of the introduction.
The second paragraph started with the same contents as the introduction, line 48.
In the third paragraph, the authors already mentioned the bioavailability of polyphenol line 56.
The beginning of the forth paragraph talked about the same thing in 3.7
Some punctuation errors:
Line 15. Space after properties
Reviewer 2 Report
Comments and Suggestions for Authors
This manuscript describes novel polyphenol modulates UCP1-dependent thermogenesis to counteract obesity exemplified by compound 18a, which was reported from a previous study. 18a induces thermogenesis mediated by UCP1 and activates brown fat thermogenesis through the AMPK-PGC-1α pathway. In addition, a series of experiments have well proved the effect of 18a in vitro and in vivo. This article does in-depth research and expands the biological activity of polyphenolic structures. As such, I recommend this manuscript to be published after appropriate revision as below:
1. The collection time for plasma concentration of 18a is too brief and limited to a single point; extending this collection period is necessary. Additionally, it would be beneficial to supplement these findings with bioavailability data (F) for 18a.
2. In the article, the target of 18a was not mentioned (your previous research target the DRAK2 protein). Can you use chemical biology methods to find targets for 18a? This contributes to the quality of the article.
3. In the introduction, it is necessary to provide an overview of the AMPK-PGC-1α pathway and discuss the interaction between AMPK or PGC-1α with UCP1.
4. The article did a lot of phenotypic experiments to verify the effect of 18a. but more mechanisms need to be added to solidify its conclusions.
5. In the literature, you will not find compound18a or 18a appearing in titles or keywords, separately.
6. There are many details in the article that need to be revised, as follows: Line15, "," symbol format; line 19, "in vivo" italics; line 36, ".Ucp1" space, line 45, "quercetin [15]" comma; line 534, " e19"deletion; line 574, 578, and 582, page. I just checked it briefly and found many errors. Please check the article carefully.
Comments on the Quality of English LanguageThe author needs to find a professional to reorganize the language of the article.
just give two sentence that have a problem:
1) line 20 to line 21, This sentence is ambiguous, rearrange it.
2) line 33 to line 35, the sentence is too long.
Round 2
Reviewer 2 Report
Comments and Suggestions for Authors
Accept in present form